# The Use of Ultrasonic Bone Scalpel (UBS) in Unilateral Biportal Endoscopic Spine Surgery (UBESS): Technical Notes and Outcomes

**DOI:** 10.3390/jcm12031180

**Published:** 2023-02-02

**Authors:** Sung Huang Laurent Tsai, Chia-Wei Chang, Tung-Yi Lin, Ying-Chih Wang, Chak-Bor Wong, Abdul Karim Ghaith, Mohammed Ali Alvi, Tsai-Sheng Fu, Mohamad Bydon

**Affiliations:** 1Department of Orthopedic Surgery, Chang Gung Memorial Hospital, Keelung Branch, Bone and Joint Research Center, Chang Gung University, F7, No 222 Mai-King Road, Keelung 20401, Taiwan; 2Mayo Clinic Neuro-Informatics Laboratory, Mayo Clinic, Rochester, MN 55902, USA; 3Department of Neurological Surgery, Mayo Clinic, Rochester, MN 55902, USA

**Keywords:** ultrasonic bone scalpel, unilateral biportal endoscopic surgery, minimally invasive surgery, spinal stenosis, decompression, technical notes

## Abstract

Study Design: Case Series and Technical Note, Objective: UBS has been extensively used in open surgery. However, the use of UBS during UBESS has not been reported in the literature. The aim of this study was to describe a new spinal surgical technique using an ultrasonic bone scalpel (UBS) during unilateral biportal endoscopic spine surgery (UBESS) and to report the preliminary results of this technique. Methods: We enrolled patients diagnosed with lumbar spinal stenosis who underwent single-level UBESS. All patients were followed up for more than 12 months. A unilateral laminotomy was performed after bilateral decompression under endoscopy. We used the UBS system after direct visualization of the target for a bone cut. We evaluated the demographic characteristics, diagnosis, operative time, and estimated blood loss of the patients. Clinical outcomes included the visual analog scale (VAS), the Oswestry Disability Index (ODI), the modified MacNab criteria, and postoperative complications. Results: A total of twenty patients (five males and fifteen females) were enrolled in this study. The mean follow-up period was 13.2 months (range 12–17 months). The VAS score, ODI, and modified MacNab criteria classification improved after the surgery. A minimal mean blood loss of 22.1 mL was noted during the operation. Only one patient experienced neuropraxia, which resolved within 2 weeks. There was no durotomy, iatrogenic pars fracture, or infection. Conclusions: In conclusion, our study represents the first report of the use of UBS during UBESS. Our findings demonstrate that this technique is safe and efficient, with improved clinical outcomes and minimal complications. These preliminary results warrant further investigation through larger clinical studies with longer follow-up periods to confirm the effectiveness of this technique in the treatment of lumbar spinal stenosis.

## 1. Introduction

Lumbar spinal stenosis is the most common indication for spine surgery in patients older than 65 years. It is estimated that over 200,000 people have lumbar spinal stenosis in the United States [1]. The traditional treatment for lumbar spinal stenosis is decompression with laminectomy [2]. In addition, a central laminectomy may be performed with bilateral medial facetectomies. Some treatments may include foraminotomy with or without spinal fusion [3]. Owing to the preservation of the surrounding soft tissues, minimally invasive techniques have been successful in reducing pain and accelerating functional recovery [4]. Full endoscopic decompression, especially unilateral biportal endoscopic spine surgery (UBESS), has been associated with a lower risk of complications such as dural tears, root injury, and instability after microscopic decompression [5,6,7]. Continuous visual control is more likely to achieve adequate and clear bone resection.

Traditional equipment used to remove spinal lesions during laminectomy includes high-speed drills and Kerrison Rongeurs. However, these instruments may cause spinal cord and nerve injuries if handled inappropriately [8]. Complications, such as nerve thermal injuries and dural tears, may occur during the procedure [9]. Ultrasonic bone scalpels (UBS) have been used for skull base, craniofacial, and oral surgeries. Recently, this technique has been introduced in spine surgery [10,11]. The scalpel helps create a precise, narrow incision in the vertebral arch for laminectomy and laminoplasty, which leaves the soft-tissue structures, such as the dura mater and nerves, intact [12]. Because the instrument has less tip-bone contact force, a self-irrigation system, and causes less vibration, it is useful for performing procedures near the dura mater and neural structures due to the decreased risk of excessive mechanical and thermal injuries [13]. Studies in the past have reported the use of an UBS in open spine surgery [14]. Moon et al. have proven the use of UBS in open lumbar decompression surgeries to be safe [15]. However, no studies have demonstrated its use in UBESS. This article introduces and describes our use of the UBS system in UBESS.

## 2. Methods

### 2.1. Technical Note

UBESS was performed in the lower lumbar region for spinal stenosis using an ultrasonic bone scalpel (Misonix BoneScalpel knife and hook type) (Figure 1). The surgical procedure was executed by a single surgeon who possessed a substantial level of proficiency in the field of general spinal surgery. However, it should be noted that the surgeon was at the nascent stage of acquiring competency in the specific technique under investigation. We included patients based on the following criteria: single-level lumbar stenosis, neurological intermittent claudication, and radiculopathy with leg pain predominantly refractory to conservative treatment for 3 months. We excluded patients with a history of spine surgery, spinal instability, spondylolisthesis, multilevel spinal stenosis, spinal infections, spinal trauma, and apparent ligament flavum calcification. In addition, only patients available for a follow-up of more than 12 months after UBESS were included. All patients underwent an MRI both preoperatively and postoperatively. The following parameters were collected: age, sex, and surgical level. Outcomes included operation time (minutes), blood loss (mL), hospital length of stay (days), and complications including recurrence, hematoma, iatrogenic pars fracture, durotomy, revision surgery, and infection. The subjective pain evaluation of the visual analog scale (VAS) was scored for the lower limb. The functional outcomes of the Oswestry Disability Index (ODI) and the modified MacNab criteria (postoperative) were documented [16,17]. We evaluated the morphology of the dural sac on T2-weighted axial MRI by using the Schizas system, a 7-grade classification system that takes into account the rootlet/CSF fluid ratio. The system categorizes the levels of stenosis into Grade A, no or minor stenosis; Grade B, moderate stenosis; Grade C, severe stenosis; and Grade D, extreme stenosis. Each grade is defined by the visibility and location of rootlets and epidural fat in the dural sac. Using this system, we were able to accurately assess the degree of stenosis in the dural sac and make an appropriate diagnosis [18]. Clinical assessment was conducted preoperatively, immediately after the operation, and at 6 weeks, 3 months, 6 months, 12 months, and the final follow-up postoperatively. The study protocol was approved by the institutional review board (IRB) at the study site, Chang Gung Memorial Hospital (IRB 202101251B0). The IRB approved the waiver of the participant’s consent.

### 2.2. Surgical Procedure

The procedures were performed under general anesthesia with the patient in a prone position. A radiolucent Wilson frame was used to decrease the pressure on the patient’s abdomen. A waterproof surgical drape was applied before the start of the procedure.

Our stenotic target level was identified through fluoroscopic guidance. The incision point was the intersection between the perpendicular midline of the lower laminar margin and the spinous process. A right-handed surgeon performed a left-sided approach on all the patients. Two 1 cm incisions were made: the cranial incision was made for the scope portal, which was used for continuous irrigation and endoscopic viewing. The other caudal incision was made for the working portal, which was used for the insertion of instruments used for decompression, including our UBS. The two portals were separated by approximately 3 cm. The study used a 30 degree angled endoscope for the UBESS due to equipment availability at the institution. The 30 degree endoscope was inserted through the scope portal after the cannula was inserted for better visualization of the operative field. A saline irrigation pump was connected to the endoscope and set to a pressure of 20 mmHg to prevent an excessive increase in the epidural pressure. Radiofrequency probes were used to control minor bleeding when identified through the endoscope. Soft-tissue remnants covering the lamina and ligamentum flavum were debrided. An ipsilateral partial laminotomy was performed under full endoscopic vision using a UBS (knife type) followed by a high-speed burr and Kerrison rongeurs. The lower border of the cranial lamina, the midline spino-laminar junction, and the upper border of the caudal lamina were removed using the UBS (hook type) (Figure 2). The endoscope was moved to the contralateral side with Kerrison punches and a curette used for contralateral decompression. If the patient had symptomatic ipsilateral herniation, a discectomy was performed. Epidural bleeding was controlled throughout the procedure, while the UBS was used for decompression. Neural decompression was confirmed by checking dural pulsation and full mobilization of the nerve root. After removing both portals, skin closure was performed using 3-0 Nylon.

### 2.3. Statistical Analysis

Descriptive statistics were computed for patient characteristics. Categorical variables were expressed as actual numbers, whereas continuous variables were represented by the mean and standard deviation. A paired sample *t*-test was calculated for the statistical comparison of continuous variables. A two-sided *p* value < 0.05 was considered statistically significant. All statistical analyses were performed using Stata MP (14.2, 1985–2015; StataCorp LLC, College Station, TX 77845, USA).

## 3. Results

The present study cohort consisted of 90 patients treated between 2019 and 2020 by a single surgeon. Ten cases with a history of previous spine surgery, 65 cases with spinal instability, spondylolisthesis, or multilevel spinal stenosis, five cases with spinal infections, seven cases with spinal trauma, and three cases with apparent ligament flavum calcification were excluded from the study. The final study cohort consisted of twenty patients, five of whom were male and fifteen of whom were female, all of whom had single-segment degenerative spinal stenotic lesions (Figure 3). Out of a total of twenty cases, three cases were classified as Grade B (moderate stenosis), fifteen cases were classified as Grade C (severe stenosis), and two cases were classified as Grade D (extreme stenosis) using the Schizas system for evaluating the morphology of the dural sac on T2-weighted axial MRI. (Table 1) All cases had symptoms of back pain and radiculopathy for more than 3 months (range: 3–18 months). Most lesions were at the L4-5 level (N = 15), and five were at L3-4. The mean follow-up period was 13.2 months. The mean age was 66.1 ± 7.8 years. The mean blood loss was 22.1 ± 5.3 mL. The mean operative time was 74.4 ± 9.4 min. The total length of hospital stay (LOS) was 2.3 ± 0.6 days. The VAS score, ODI score, and modified MacNab criteria significantly improved after the surgery. The VAS scores improved from 7.7 ± 0.2 to 2.8 ± 0.2 immediately after the operation (*p* < 0.05), and the ODI scores improved from 64.3 ± 1.3 to 23.8 ± 1.5 (*p* < 0.05) (Figure 4 and Figure 5). The modified MacNab criteria were reported as excellent in seventeen patients, good in two, and fair in one. The UBS system provided clear, full visualization throughout the UBESS procedure. Additional discectomies were done in five patients. After the procedure, only one patient experienced neuropraxia, which resolved spontaneously within two weeks. No iatrogenic pars fractures, durotomies, or infections were reported. None of these cases required revision surgery. MRI follow-up revealed full decompression of the single-level lesion (Figure 6). The full procedure can be accessed from the Appendix A.

## 4. Discussion

In the present study, we demonstrated the use of the UBS system in UBESS to achieve maximal preservation of normal musculoligamentous structures [19]. At the same time, UBESS can achieve full direct visualization during the procedure [20]. UBS allows en bloc bone dissection with the knife type and precise bone ablation with the hook type [21]. The features of UBS can be summarized as the “triple S”. The first S stands for “Strong”; the UBS is a safe, fast, and precise osteotome during UBESS [10,11]. The second S stands for “Safe”; the UBS spares soft tissue to possibly prevent nerve injuries. The third S stands for “Speed”. Reports claim that the UBS decreases the risk of mechanical injury, produces less heat, and reduces osseous bleeding. These facts allow for clear visualization of the surgical field and a shorter operative time [9,22]. This also decreases complications such as durotomy, nerve injuries, and hematoma. Previous articles reported that UBS is associated with a dural tear rate of 0.9–6.5% and a neural damage rate of 0.2–3.2% during open surgery [9,12,22]. Liang et al. reported that the most common complications during UBESS were durotomies and hematomas, with incidence rates of 2% and 1%, respectively. Our preliminary findings showed no durotomies, nerve damage, or hematoma during bone removal. One patient had neuropraxia, which resolved spontaneously within 2 weeks. However, bed rest may spontaneously resolve an intraoperative dural tear that cannot be repaired [23]. Performing open microscopic surgery is the best option for better visualization and repairing a large dural tear [24]. Kim et al. proposed that cotton buffers can be used to repair large dural tears and to prevent direct vertical force to the spinal cord when using Kerrison Rongeurs [22].

Complications such as hematoma can be eliminated due to the decrease in bleeding during the operation. The self-irrigation feature of the UBS system decreases blood loss by decreasing the overall operation time. The use of UBS has been shown to be safe and faster for cutting when compared to the high-speed burr [10]. However, a postoperative epidural hematoma may occur if coagulation is not achieved. The UBS instrument can distinguish bone from soft tissue. The scalpel tip purchases bone with the oscillation of the ultrasound and spares the soft-tissue structures with reflexive vibrations [11]. The UBS system also decreased blood loss and operative time, which in turn resulted in a lower rate of surgical site infection and wound dehiscence [10]. Faster mobilization following the operation is achieved when a catheter is less likely to be needed for hematoma draining.

During UBESS, two unilateral entry points are made on the same side of the operator. One is used for the endoscope, and the other is used as an entry point for other instruments, including the UBS [25]. Compared with the uniportal endoscopic approach, the working portal is used only for surgical instruments. Therefore, controlling and moving the instruments is unrestricted and smooth [19]. This can decrease the risk of instrumental competition, which might cause structural damage during the procedure [23].

In this paper, we describe the first reported use of UBS in the UBESS. This approach combines the benefits of both direct visualization and reduced soft tissue disruption. The results of our study showed that the use of UBS during UBESS led to minimal blood loss, rapid recovery, and a shortened hospital stay. While the use of UBS during UBESS has several advantages, it is important to acknowledge its limitations as well. One major limitation of this study is that the findings are based on the experience of a single surgeon at a single large medical center. Therefore, the results may not be generalizable to other patient populations or to other surgical teams. It is important to note that further studies with larger sample sizes and involving multiple surgeons from different institutions are needed to confirm the safety and efficacy of UBS during UBESS. Additionally, long-term follow-up and outcomes are necessary to fully evaluate the potential benefits of this technique.

Future directions of this research include performing larger and more comprehensive studies to validate the safety and efficacy of UBS in UBESS. Additionally, the use of UBS in other spinal conditions and procedures should also be explored. Furthermore, the combination of UBS with other advanced technologies such as navigation, intraoperative monitoring, and robotic assistance should be studied to further enhance the precision, safety, and efficiency of spinal surgery. Overall, UBS in UBESS is a promising new technique that has the potential to improve the outcomes of patients undergoing spinal surgery. The results of our study provide a foundation for future research and clinical applications of this innovative technology.

## 5. Conclusions

The purpose of this technical note is to present preliminary evidence of the feasibility of utilizing an ultrasonic bone scalpel (UBS) during unilateral biportal endoscopic spine surgery (UBESS). While our experience with this technique is limited, our preliminary findings suggest that it is a safe and viable option. However, it is important to note that further investigation is necessary to fully determine the feasibility of UBS in UBESS and to identify any potential contraindications for the procedure. The current study provides a foundation for future research to further evaluate the safety and efficacy of using UBS in UBESS.

## Figures and Tables

**Figure 1 jcm-12-01180-f001:**
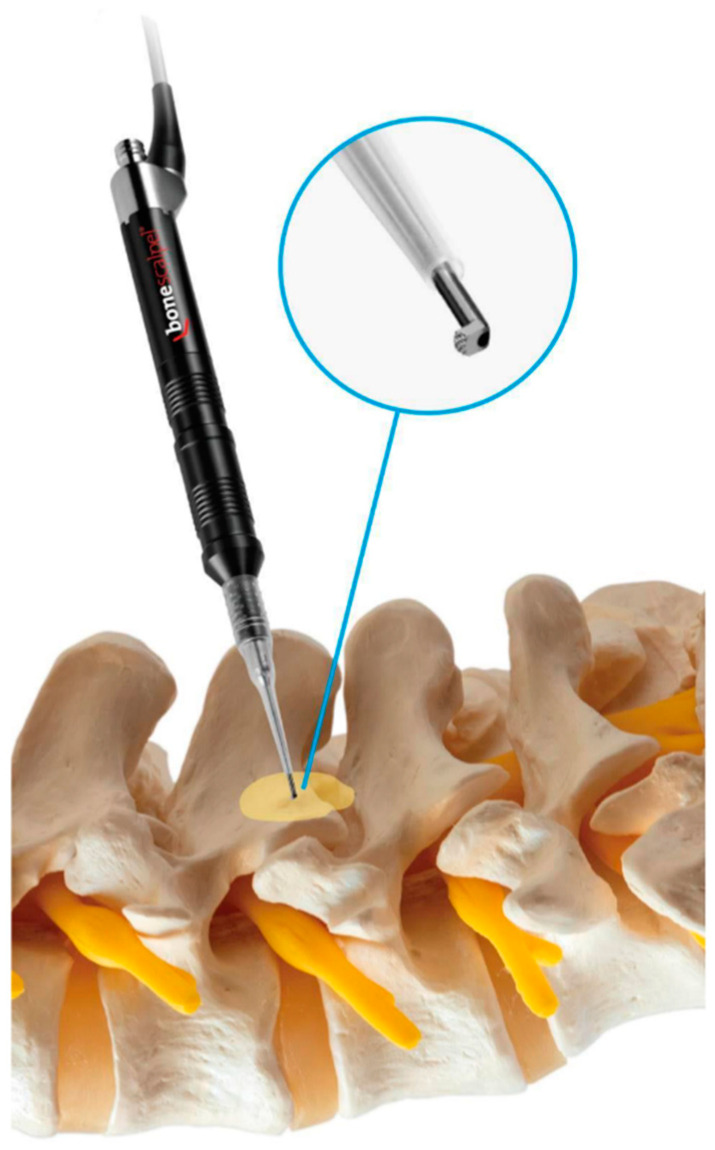
Illustration showing the use of the ultrasonic bone scalpel (UBS) hook type during single-level spinal stenosis decompression. The UBS system provides a strong, safe, and speedy cut during the operation.

**Figure 2 jcm-12-01180-f002:**
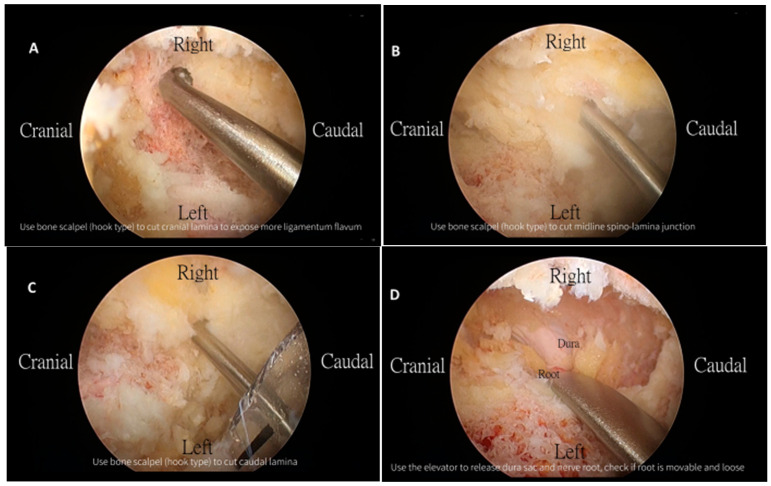
Intraoperative images of the technical use of UBS for decompression. (**A**). The use of the UBS hook type to cut the cranial lamina. (**B**). The use of the UBS hook type to cut the midline spinolaminar junction. (**C**). The use of the UBS hook type to cut the caudal lamina. (**D**). After decompression, we used the elevator to separate the dural sac and nerve root to check if the root was movable and loose.

**Figure 3 jcm-12-01180-f003:**
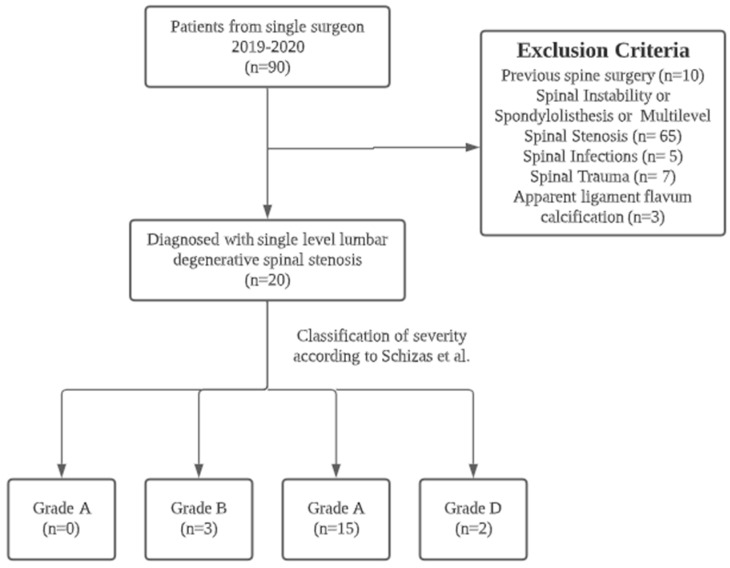
Patient flowchart for inclusion and exclusion selections.

**Figure 4 jcm-12-01180-f004:**
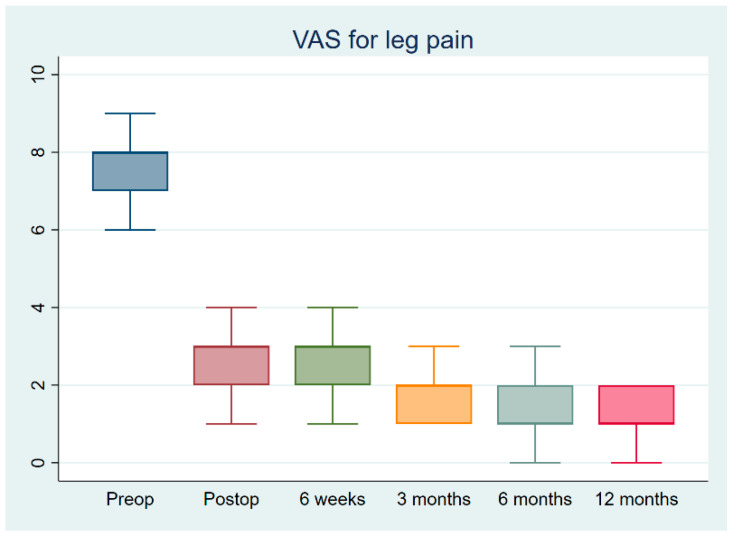
Visual Analog Scale (VAS) for leg pain demonstrates significant improvement from before to immediately after surgery.

**Figure 5 jcm-12-01180-f005:**
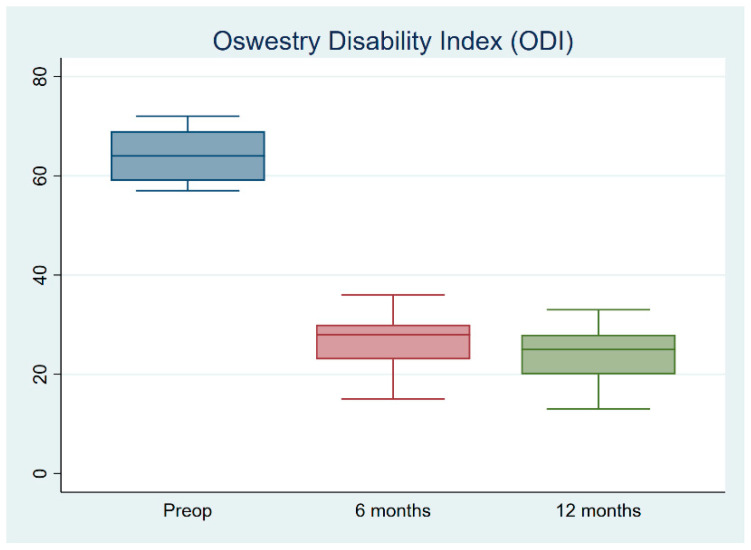
The Oswestry Disability Index (ODI) significantly improved 6 months after surgery.

**Figure 6 jcm-12-01180-f006:**
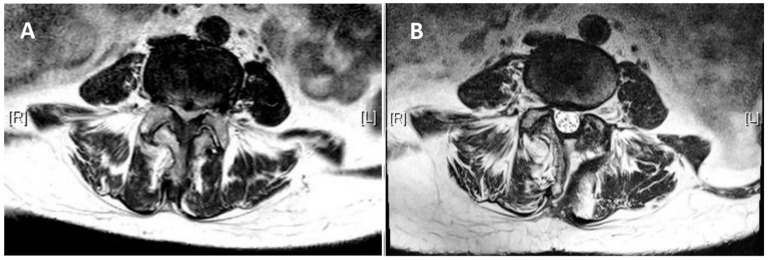
MRI follow-up at 12 months for a 61-year-old female with a diagnosis of L4-5 spinal stenosis. (**A**). A preoperative MRI shows a severe spinal stenotic lesion at L4-5. (**B**). Postoperative MRI showing full decompression of the spinal canal.

**Table 1 jcm-12-01180-t001:** Patient demographics and surgical and hospital characteristics.

Factors	Value
Age (mean, SD)	66.1 (7.8)
Sex (M/F)	5/15
Level (N)	
L4-5	15
L3-4	5
Follow-up (months)	13.2 (0.71)
Mean operative time (mins)	74.4 (9.4)
Blood loss (mL)	22.1 (5.3)
Length of stay (days)	2.3 (0.62)
Schizas Score (N) *	
Grade A	0
Grade B	3
Grade C	15
Grade D	2
Modified MacNab Criteria (N)	
Excellent	17
Good	2
Fair	1
Poor	0

* Grade A: no or minor stenosis; Grade B: moderate stenosis; Grade C: severe stenosis; and Grade D: extreme stenosis.

## Data Availability

Not applicable.

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
