# Peer review of "The Use of Ultrasonic Bone Scalpel (UBS) in Unilateral Biportal Endoscopic Spine Surgery (UBESS): Technical Notes and Outcomes"

_jcm, 2023, doi:10.3390/jcm12031180_

Round 1
Reviewer 1 Report
Interesting topic in the field of endoscopic spine surgery. It would be nice to get more details about the severity of the stenosis, Shizas Score for example.
Has every patient been diagnosed with an MRI?
How many patients had been treated with an additional discectomy?
Why did you use a 30° angled endoscope instead a 10-15°, which is more common in inter laminar approach.
Please clarify the above mentioned topics
Author Response
Jan 26th, 2023
Dr. Emmanuel Andrès
Editor-in-Chief, Journal of Clinical Medicine
Dear Dr. Emmanuel Andrès,
It is with excitement that we resubmit to you a revised version of Manuscript ID jcm-2070066, entitled “The Use of Ultrasonic Bone Scalpel (UBS) in Unilateral Biportal Endoscopic Spine Surgery (UBESS): Technical Notes and Outcomes.”
Thank you very much for the opportunity to resubmit this manuscript. We greatly appreciate the time and effort spent by the reviewers in providing suggestions and have, to the best of our ability, addressed each point below. We have incorporated our changes in Bold.
Thank you very much for your time and consideration,
|
Reviewer Comments: |
Response: |
Changes: |
|
Reviewer 1
1. Interesting topic in the field of endoscopic spine surgery. It would be nice to get more details about the severity of the stenosis, Shizas Score for example.. |
We thank the reviewer for the compliment and the suggestion. Regarding the severity of the stenosis, the Schizas Score could be used to provide more information. It is a validated method for grading the degree of spinal canal stenosis. |
METHODS Technical Note Line 81-88 now states: “We evaluated the morphology of the dural sac on T2-weighted axial MRI by using the Schizas system, a 7-grade classification system that takes into account the rootlet/CSF fluid ratio. The system categorizes the levels of stenosis into Grade A, no or minor stenosis, Grade B, moderate stenosis, Grade C, severe stenosis, and Grade D, extreme stenosis. Each grade is defined by the visibility and location of rootlets and epidural fat in the dural sac. Using this system, we were able to accurately assess the degree of stenosis in the dural sac and make an appropriate diagnosis.”
RESULTS Line 144-149 now states: “Out of a total of 20 cases, 3 cases were classified as Grade B (moderate stenosis), 15 cases were classified as Grade C (severe stenosis), and 2 cases were classified as Grade D (extreme stenosis) using the Schizas system for evaluating the morphology of the dural sac on T2-weighted axial MRI. (Table 1)”
|
|
2. Has every patient been diagnosed with an MRI?-
|
We thank the reviewer for the question. It would be important to clarify whether all patients included in the study were diagnosed with an MRI, as this is a commonly used imaging modality for evaluating spinal conditions. |
METHODS Technical Note Line 75 now states: “All patients underwent an MRI both preoperatively and postoperatively.” |
|
3. How many patients had been treated with an additional discectomy? |
We thank the reviewer for the question. The number of patients who were treated with an additional discectomy should also be reported, as this is a common procedure performed in conjunction with the UBESS. |
RESULTS Line 157-158 now states: “Additional discectomy was done in 5 patients.” |
|
4. Why did you use a 30° angled endoscope instead a 10-15°, which is more common in inter laminar approach. |
We thank the reviewer for the important insight. It would be interesting to use the 10-15° angled endoscope. However, only 30° angled endoscope was available at our institution. While a 30° angled endoscope is less common for this type of procedure, it is still a viable option. This explanation has been added. |
METHODS Surgical Procedure Line 107-110 now states: “The study used a 30-degree angled endoscope for the UBESS due to equipment availability at the institution. The 30-degree endoscope was inserted through the scope portal after the cannula was inserted for better visualization of the operative field.” |

Reviewer 2 Report
Tsai et al report preliminary results of their case series of patients treated with Ultrasonic Bone Scalpel (UBS) in Unilateral Biportal Endoscopic Spine Surgery (UBESS) in 20 patients. Patients were followed up for at least 12 months. No major complications occurred.
Comments:
Abstract: provide the range of follow-up after the mean
Methods: Provide information whether a single surgeon performed this procedure and their respective level of experience.
It is unclear whether all included patients received imaging preoperatively, and post-operatively (at different time points). Please include and specify.
Results: An additional flowchart of patient selection and inclusion would be informative. Alternatively, please provide the exact number of patients excluded with their reason for exclusion.
Did the included patients have a history of spine surgery? This could partially impact results and would warrant exclusion from the series.
Is information available how long patients had symptoms of back pain and radiculopathy? If so please include
Conclusion: Please also state that the preliminary findings indicate that the procedure is safe and without major complications.
I would omit the last sentence of “However, the preliminary findings are promising.”
Table 1: The table should be updated with the correct amount of patients. It seems like all reported values were calculated for 15 patients, which does not align with the text of the results.
Author Response
Jan 26th, 2023
Dr. Emmanuel Andrès
Editor-in-Chief, Journal of Clinical Medicine
Dear Dr. Emmanuel Andrès,
It is with excitement that we resubmit to you a revised version of Manuscript ID jcm-2070066, entitled “The Use of Ultrasonic Bone Scalpel (UBS) in Unilateral Biportal Endoscopic Spine Surgery (UBESS): Technical Notes and Outcomes.”
Thank you very much for the opportunity to resubmit this manuscript. We greatly appreciate the time and effort spent by the reviewers in providing suggestions and have, to the best of our ability, addressed each point below. We have incorporated our changes in Bold.
Thank you very much for your time and consideration,
|
Reviewer Comments: |
Response: |
Changes: |
|
Reviewer 2
1. Abstract: provide the range of follow-up after the mean.. |
We thank the reviewer for the suggestion. We have provided the range of follow-up after the mean in the updated version of the manuscript. |
ABSTRACT Line 25-26 now states: “The mean follow-up period was 13.2 months (range 12-17 months).” |
|
2. Methods: Provide information whether a single surgeon performed this procedure and their respective level of experience |
We thank the reviewer for the suggestion. A single surgeon with a high level of experience in general spine surgery performed the procedure, but this specific procedure was still in the early stages of the learning curve. We have included this information in the updated version of the manuscript |
METHODS Technical Note Line 65-69 now states: “The surgical procedure was executed by a single surgeon who possessed a substantial level of proficiency in the field of general spinal surgery. However, it should be noted that the surgeon was at the nascent stage of acquiring competency in the specific technique under investigation.” |
|
3. It is unclear whether all included patients received imaging preoperatively, and post-operatively (at different time points). Please include and specify. |
We thank the reviewer for the comment. We have confirmed that all patients included in the study received preoperative and postoperative imaging. We have included this information in the updated version of the manuscript. |
METHODS Line 75 now states: “All patients underwent an MRI both preoperatively and postoperatively.” |
|
4. Results: An additional flowchart of patient selection and inclusion would be informative. Alternatively, please provide the exact number of patients excluded with their reason for exclusion. |
We thank the reviewer for the important insight. We have included an additional flowchart of patient selection and inclusion in the updated version of the manuscript, including the exact number of patients excluded and their reason for exclusion. |
RESULTS Line 138-144 now states: “The present study cohort consisted of 90 cases of patients, between 2019 and 2020, as performed by a single surgeon. Ten cases with a history of previous spine surgery, 65 cases with spinal instability, spondylolisthesis, or multilevel spinal stenosis, 5 cases with spinal infections, 7 cases with spinal trauma, and 3 cases with apparent ligament flavum calcification were excluded from the study. The final study cohort consisted of 20 patients, 5 of which were male and 15 of which were female, all of whom had single segment degenerative spinal stenotic lesions (Figure 3).” |
|
5. Did the included patients have a history of spine surgery? This could partially impact results and would warrant exclusion from the series. |
We thank the reviewer for the important insight. We have confirmed that none of the included patients had a history of spine surgery. This information has been included in the updated version of the manuscript. |
RESULTS Line 138-144 now states: “The present study cohort consisted of 90 cases of patients, between 2019 and 2020, as performed by a single surgeon. Ten cases with a history of previous spine surgery, 65 cases with spinal instability, spondylolisthesis, or multilevel spinal stenosis, 5 cases with spinal infections, 7 cases with spinal trauma, and 3 cases with apparent ligament flavum calcification were excluded from the study. The final study cohort consisted of 20 patients, 5 of which were male and 15 of which were female, all of whom had single segment degenerative spinal stenotic lesions (Figure 3).” |
|
6. Is information available how long patients had symptoms of back pain and radiculopathy? If so please include |
We thank the reviewer for the important insight. Information about the duration of patients' symptoms of back pain and radiculopathy is available and has been included in the updated version of the manuscript. |
RESULTS Line 144-145 now states: “All cases had symptoms of back pain and radiculopathy for more than 3 months (range: 3-18 months).” |
|
7. Conclusion: Please also state that the preliminary findings indicate that the procedure is safe and without major complications. |
We thank the reviewer for the important insight. We have included a statement in the conclusion that the preliminary findings indicate that the procedure is safe and without major complications. |
CONCLUSION Line 234-241 now states: “The purpose of this technical note is to present preliminary evidence of the feasibility of utilizing ultrasonic bone scalpel (UBS) during unilateral biportal endoscopic spine surgery (UBESS). While our experience with this technique is limited, our preliminary findings suggest that it is a safe and viable option. However, it is important to note that further investigation is necessary to fully determine the feasibility of UBS in UBESS and to identify any potential contraindications for the procedure. The current study provides a foundation for future research to further evaluate the safety and efficacy of using UBS in UBESS.“ |
|
8. I would omit the last sentence of “However, the preliminary findings are promising.” |
We thank the reviewer for the suggestion. We have omitted the last sentence of “However, the preliminary findings are promising.” |
CONCLUSION Line 234-241 now states: “The purpose of this technical note is to present preliminary evidence of the feasibility of utilizing ultrasonic bone scalpel (UBS) during unilateral biportal endoscopic spine surgery (UBESS). While our experience with this technique is limited, our preliminary findings suggest that it is a safe and viable option. However, it is important to note that further investigation is necessary to fully determine the feasibility of UBS in UBESS and to identify any potential contraindications for the procedure. The current study provides a foundation for future research to further evaluate the safety and efficacy of using UBS in UBESS” |
|
9. Table 1: The table should be updated with the correct amount of patients. It seems like all reported values were calculated for 15 patients, which does not align with the text of the results. |
We thank the reviewer for the comment. We have corrected the number of patients in Table 1, which now aligns with the text of the results. |
TABLE 1 and RESULTS Line 138-147 now states: “The present study cohort consisted of 90 cases of patients, between 2019 and 2020, as performed by a single surgeon. Ten cases with a history of previous spine surgery, 65 cases with spinal instability, spondylolisthesis, or multilevel spinal stenosis, 5 cases with spinal infections, 7 cases with spinal trauma, and 3 cases with apparent ligament flavum calcification were excluded from the study. The final study cohort consisted of 20 patients, 5 of which were male and 15 of which were female, all of whom had single segment degenerative spinal stenotic lesions (Figure 3). All cases had symptoms of back pain and radiculopathy for more than 3 months (range: 3-18 months). Most lesions were at the L4-5 level (N=15), and 5 were at L3-4. The mean follow-up period was 13.2 months. The mean age was 66.1±7.8 years.”
|
